# How Light Reactions of Photosynthesis in C4 Plants Are Optimized and Protected under High Light Conditions

**DOI:** 10.3390/ijms23073626

**Published:** 2022-03-26

**Authors:** Wioleta Wasilewska-Dębowska, Maksymilian Zienkiewicz, Anna Drozak

**Affiliations:** Department of Molecular Plant Physiology, Institute of Environmental Biology, Faculty of Biology, University of Warsaw, I. Miecznikowa 1, 02-096 Warsaw, Poland; wp.wasilewska@uw.edu.pl (W.W.-D.); mm.zienkiewicz@uw.edu.pl (M.Z.)

**Keywords:** bundle sheath chloroplasts, C4 photosynthesis, cyclic electron transport, environmental factors, high light intensity, mesophyll chloroplasts, NAD-ME, NADP-ME, PEPCK subtypes of C4 photosynthesis, xanthophyll cycle

## Abstract

Most C4 plants that naturally occur in tropical or subtropical climates, in high light environments, had to evolve a series of adaptations of photosynthesis that allowed them to grow under these conditions. In this review, we summarize mechanisms that ensure the balancing of energy distribution, counteract photoinhibition, and allow the dissipation of excess light energy. They secure effective electron transport in light reactions of photosynthesis, which will lead to the production of NADPH and ATP. Furthermore, a higher content of the cyclic electron transport components and an increase in ATP production are observed, which is necessary for the metabolism of C4 for effective assimilation of CO_2_. Most of the data are provided by studies of the genus *Flaveria*, where species belonging to different metabolic subtypes and intermediate forms between C3 and C4 are present. All described mechanisms that function in mesophyll and bundle sheath chloroplasts, into which photosynthetic reactions are divided, may differ in metabolic subtypes as a result of the different organization of thylakoid membranes, as well as the different demand for ATP and NADPH. This indicates that C4 plants have plasticity in the utilization of pathways in which efficient use and dissipation of excitation energy are realized.

## 1. Diversity of C4 Photosynthesis

Photosynthesis is a complex metabolic process in which solar energy is utilized to convert atmospheric carbon dioxide (CO_2_) into organic compounds. Traditionally, it is divided into two phases: (1) the light energy captured by the pigment–protein complexes is exchanged into energy-rich bonds of molecules such as ATP and NADPH, and (2) when ATP and NADPH drive the fixation of CO_2_ in carbohydrates. Approximately 85% of all higher plants use the C3 photosynthetic pathway [1]. In this process, the first step of CO_2_ fixation is catalyzed by ribulose-1,5-bisphosphate carboxylase/oxygenase (RuBisCO) and two three-carbon molecules (3-phosphoglyceric acid) are produced by carboxylation of ribulose-1,5-bisphosphate. However, RuBisCO also shows oxygenase activity, which leads to carbon loss by photorespiration. Plants have evolved mechanisms that allow them to concentrate CO_2_ at the RuBisCO site and thus eliminate or reduce the oxygenase activity of RuBisCO. One of them is C4 photosynthesis, a pathway described in the 1960s [2,3,4]. In this type of photosynthesis (Figure 1A), the assimilation and reduction of CO_2_ are spatially separated and catalyzed by two different enzymes. Carbon dioxide assimilation occurs in the cytoplasm of mesophyll (M) cells, where CO_2_ is initially converted to HCO_3_^−^, which is then incorporated into phosphoenolpyruvate (PEP) to form oxaloacetate (OAA) by phosphoenolpyruvate carboxylase (PEPC). Then, OAA is converted to malate or aspartate, and these carboxylic acids are transported to bundle sheath (BS) cells, where they are decarboxylated. The CO_2_ released is then assimilated by RuBisCO (present only in the chloroplasts of BS cells) through the Calvin–Benson cycle [5]. Due to differences in a type of carboxylic acid transferred to BS cells, and the main enzyme responsible for the decarboxylation and regeneration of the CO_2_ acceptor in M cells, three biochemical subtypes of C4 photosynthesis are recognized: NADP-ME, NAD-ME, and PEPCK or PCK [5,6,7]. In the most common NADP-ME subtype, malate is transferred from the mesophyll to bundle sheath cells and CO_2_ is released into the chloroplasts of BS by the NADP-dependent malic enzyme (NADP-ME). However, in the NAD-ME subtype, aspartate is the main carboxylic acid that is transported from M to BS cells, and decarboxylation is catalyzed by mitochondrial NAD-dependent malic enzyme (NAD-ME). In the third PEPCK subtype, both aspartate and malate are transferred to BS cells, where aspartate is converted to OAA in the cytosol. Then, it is decarboxylated by PEP carboxykinase (PEPCK or PCK). Moreover, malate synthesized in the mesophyll chloroplast is transported into BS mitochondria, where decarboxylation catalyzed by NAD-ME occurs. Furthermore, although C4 photosynthesis has been classified into three subtypes, accumulating evidence indicates that many C4 plants use a combination of organic acids and decarboxylases during CO_2_ fixation, and the C4-type categories are not rigid [8]. The C4 metabolic cycle consumes more energy than C3 photosynthesis. Fixation of 1 CO_2_ requires 5 ATP and 2 NADPH for plants driving the NADP-ME and NAD-ME photosynthesis subtypes and 3.6 ATP and 2.3 NADPH for the PEPCK subtype (Figure 1B), whereas for the C3 cycle this energy demand is 3 ATP and 2 NADPH [9,10].

Separation of assimilation and CO_2_ reduction into two cell types involves several adaptations in the anatomy and ultrastructure of the leaf [5]. This special leaf anatomy in the C4 plant is called Kranz and characterizes thick-walled (containing suberin) large parenchyma cells (BS cells) that tightly surround vascular bundles, while mesophyll cells, with much thinner walls, are located between the leaf epidermis and BS cells [11]. In addition, mesophyll and BS cells are connected by a dense network of plasmodesmata through which an intensive transport of metabolites takes place [5]. Furthermore, the M and BS chloroplasts in species belonging to different subtypes of C4 photosynthesis exhibit structural dimorphism. In the NADP-ME subtype, chloroplasts localized in M cells have grana, whereas BS chloroplasts are deficient in these structures or grana stacks are rare [12,13]. In contrast, in the NAD-ME type, mesophyll chloroplasts are more deficient in grana than the BS chloroplast [14,15]. In the PEPCK subtype, M and BS chloroplasts demonstrate a similar pattern of granal development [13,16]. This dimorphism is related to differences in the need for NADPH and ATP between the chloroplasts of the mesophyll and bundle sheath cells to support C4 photosynthesis [9].

This spatial separation of CO_2_ assimilation and reduction allows for up to a tenfold increase in the CO_2_ concentration in BS cells compared to the natural concentration of this gas in the air [17]. The maintenance of the high concentration of CO_2_ at the RuBisCO site is also related in part to the lack or low levels of carbonic anhydrase. This prevents a rapid conversion of carbon dioxide into a carbonate anion, which is not the substrate for RuBisCO [18]. The mechanism of CO_2_ concentration, the higher CO_2_ assimilation capacity, and the reduction of stomatal conductance help C4 plants maintain higher rates of carbon gain compared to C3 plants [19,20], 50–300% higher water use efficiency [20], and higher nitrogen use efficiency [21,22]. These advantages allow C4 plants to grow at high light intensities, high temperatures, and under arid conditions, where the photorespiration process can reduce C3 photosynthesis by up to 30% [23].

Photosynthesis of C4 is a relatively recent innovation that has evolved independently more than sixty times during the last 30 million years of land plant evolution [24]. The significant differences in anatomy, biochemistry, and physiology between different C4 species are based on various genetic changes that include simple modifications of the molecular sequence of genes involved in photosynthesis, alterations of regulatory elements, gene duplications, or subcellular retargeting of proteins, and lateral transfer of genetic material. A good example of simple molecular sequence modification is PEP carboxylase, in which 21 different amino acid positions have been elucidated in grasses [25]. Larger changes in DNA sequences are also frequent and include duplication of entire genomes or genes involved in the C4 cycle [25,26,27]. Interestingly, it is currently thought that the duplication of genes encoding proteins of the core C4 cycle generally occurred before the origin of C4 photosynthesis and this may have been a contributing factor that allowed the evolution of C4 plants through the acquisition of new functions by duplicated genes. However, genetic modifications related to the evolution of C4 photosynthesis are not fully understood and are still intensively studied.

## 2. Ways of Light Energy Utilization: Balanced Distribution between Photosystems and Emission of Excess Energy as a Heat

Environmental factors, especially light intensity and quality during growth, can cause changes in pigment–protein complexes involved in the light reactions of photosynthesis. In some cases, light can cause photoinhibition, which is associated with the overproduction of reactive oxygen species (ROS) and damage of thylakoid membrane components [28]. The plant response may be more complex when there are several environmental factors, which in the case of C4 plants is quite natural due to their ability to grow in hot, and dry conditions with high light intensity [29,30].

The light energy that has been absorbed by chlorophyll molecules can be used in three ways: it can be used for photochemical reactions, its excess can be dissipated as heat, or it can be transferred as light, i.e., fluorescence (Figure 2). These processes are interdependent, so limiting photochemical reactions will result in an increase in the other two parameters [31]. As can be seen in the figure below, only a small part of the absorbed light energy is used for the course of photochemical reactions. Most of it is dissipated as heat and only 3–5% of the absorbed energy is emitted by fluorescence, but these values may change depending on environmental conditions and stress factors, including different intensities or qualities of light. Measurements of chlorophyll *a* fluorescence are commonly performed to investigate the influence of environmental factors on the photosynthetic apparatus. They provide a lot of valuable information on how much light energy is used in photochemical reactions and how much is lost (dissipated) as heat.

In this review, we focus mainly on the mechanisms that allow the adjustment of light reactions of photosynthesis to changing light conditions, including high light intensities. Unfortunately, the response of C4 photosynthetic apparatus to changing light intensities has been barely studied at the level of gene expression, and thus this review focused mainly on the role of post-translational modifications.

Changes in light intensity under natural conditions occur rapidly, and the amount of light absorbed often exceeds the efficiency of photosynthetic reactions. It should be noted that photosynthetic light reactions play a crucial role due to the products of linear (LET) and cyclic electron transport (CET), that is, the supply of ATP and NADPH, necessary for the assimilation of CO_2_ in the Calvin cycle [32]. Mechanisms that function as safety valves in chloroplasts can be divided into those universal and concern not only the protection of C4 plants, but also C3, such as the xanthophyll cycle, heat dissipation, or state transitions, and those more characteristic for C4, such as promoting cyclic electron transport and alternative CET pathways, routes to increase ATP production [32]. It should be noted that in C4 plants the response to environmental factors can be more complicated due to the existence of different metabolic subtypes (NADP-ME, NAD-ME, PEPCK), different structures of M and BS chloroplasts within these subtypes, and different requirements for ATP and NADPH.

### 2.1. Elevation of Cyclic Electron Transport Components

C4 plants have higher cyclic electron transport activity compared to C3, and this increase may have been most important for the production of the additional ATP pool required for C4 metabolism. Studies on the genus *Flaveria* showed an increase in the content of cyclic electron transport components, such as PGR5 (proton gradient regulation 5), PGRL1 (proton gradient regulation like 1) proteins, and the NDH (NADH dehydrogenase-like) complex in the chloroplast of BS cells during the evolution of C4 photosynthesis (Figure 3 and Figure 4). Furthermore, the reduction in the amount of grana or the presence of agranal thylakoid organization, as in the maize BS chloroplasts, was associated with a higher content of components such as PGR5, PGRL1, and NDH. In C4 plants, two extra ATP molecules are required to assimilate each CO_2_ molecule in the Calvin cycle. As mentioned above (Figure 1), in the NADP-ME subtype, the demand for ATP is higher in chloroplasts of BS than in mesophyll chloroplasts. It is indicated that additional ATP molecules are produced by the cyclic electron transport, which dominates in this type of chloroplast and ensures the generation of a pH gradient across the membrane thylakoid without NADPH production. Moreover, a reduction in the number of grana is associated with a lower PSII content and activity, and this down-regulation of PSII and the promotion of cyclic electron transport are related to C4 metabolism. The higher activity and content of the cyclic electron transport components were confirmed, among others, by electrophoretic methods, immunodetection, fluorescence, as well as thermoluminescence [33].

There are two key pathways in the cyclic electron transport in plants (Figure 3). One of them is classic transport with the participation of PGR5-PGRL1 proteins, the other is an alternative route involving the components of the NDH complex. The PGR5–PGRL1 complex, together with the cytochrome *b_6_f* complex, is involved in electron transport from ferredoxin to plastoquinone (PQ) [34]. The NDH complex, which consists of at least 29 subunits, also donates electrons from ferredoxin to plastoquinone. Using different electrophoretic methods, Darie et al. [35] showed that the molecular mass of the NDH complex is 550 kDa, in both M and BS maize chloroplasts. Additionally, it may exist in monomeric (550 kDa) or dimeric form (1000–1100 kDa). Moreover, they indicate that if the structure of the NDH complex is the same in two different types of maize chloroplasts, it will be very similar in all other cases [35], and only its content in thylakoid membranes may be different.

An increase in the number of individual subunits of the NDH complex is observed in BS chloroplasts of NADP-ME C4 plants, including *Zea mays*, *Sorghum bicolor,* and *Flaveria* C4 species [33,36,37,38] (Figure 4). In C4 plants, the content of the NDH complex has been shown to be four times higher in the M chloroplasts and even fourteen times higher in the BS chloroplasts compared to C3 plants [33]. Furthermore, in C3 plants, the amount of NDH complex is low and can consist of approximately 4% of total PSI, whereas in BS chloroplasts of C4 plants it can even be 40% [39]. In maize, a higher content of such subunits as NdhH, NdhK, NdhJ, and NdhE was observed, and the proportion of the NDH complex in BS and M chloroplasts was 3:1 [35].

**Figure 4 ijms-23-03626-f004:**
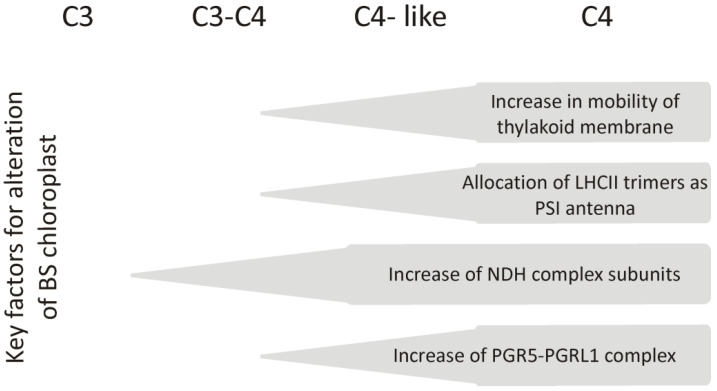
The scheme of alterations that take place in bundle sheath chloroplasts (BS) during the evolutionary process from C3 to C4 in the *Flaveria* genus. As described by Munekage and Taniguchi [38], in species representing C4 photosynthesis, the content of complexes involved in cyclic electron transport increased and the allocation of LHCII trimers as the PSI antenna occurred due to changes in the organization of thylakoid membranes.

When comparing various species representing different subtypes of C4 photosynthesis, a higher NDH content was observed in M chloroplasts in the NAD-ME subtype and in BS chloroplasts in the NADP-ME subtype, i.e., in these chloroplasts that have a higher demand for ATP. This confirms the hypothesis that NDH-dependent cyclic electron transfer plays a key role in providing the ATP needed to drive the CO_2_ concentrating mechanism [39]. It is speculated that the increase in the concentration of the NDH complex was caused by the accumulation of NADPH, resulting from a higher need for ATP than NADPH during C4 photosynthesis. In the NADP-ME subtype, NADPH is released during decarboxylation by the NADP-malic enzyme, leading to an increase in its amount in the BS chloroplast [33]. Furthermore, it has been shown that the NDH complex can participate in the dark reduction of the plastoquinone pool. Reduced plastoquinone is a signal that activates Stn7 kinase, which phosphorylates LHCII antenna systems. Presumably, the NDH complex and its functioning can maintain a certain pool of antenna phosphorylated under all conditions, allowing an even distribution of incoming light energy, efficient cyclic transport of electrons, and ATP production [40], as it occurs in chloroplasts of the maize bundle sheath cells with permanent state 2 (explained below) [41].

ATP synthase, which is responsible for the synthesis of ATP, has a localization similar to that of the complexes involved in the cyclic electron transport, in the marginal regions and stroma lamellae. The elevation in the content of the CET components is correlated with the amount of ATP synthase, especially under conditions of high light intensities, as demonstrated by Romanowska et al. [42]. The authors investigated the effect of light on the level of the CF1-α subunit in the chloroplast of M and BS cells in maize (*Zea mays*, type NADP-ME), millet (*Panicum miliaceum*, type NAD-ME), and guinea grass (*Megathyrsus maximus*, type PEPCK). They showed that light intensity had an impact on the amount of α subunit of ATP synthase, and the accumulation of additional isoform ά might have a protective role for C4 plants under high light to prevent the thylakoid lumen from overprotonation and from PSII photooxidative damage [42].

### 2.2. Function of PTOX Protein and Chlororespiration

In addition to transmembrane complexes, such as PSI, cytochrome *b_6_f*, or PSII, there are also additional components that are important in electron transport. These include the NADH chloroplast dehydrogenase complex (described above) and the plastid terminal oxidase (PTOX) [43] (Figure 3). The approximately 40 kDa PTOX protein is similar to mitochondrial alternative oxidase (AOX) and is a plastoquinone:O_2_ oxidoreductase, commonly found in plants. It is located on the stromal side of the thylakoid membrane between PSII and the cytochrome *b_6_f* complex [43]. It oxidizes the plastoquinone pool and, by donating electrons to oxygen, leads to the formation of a water molecule. This protects the plastoquinone against excessive reduction [44], for example, under conditions where the consumption of the reduced form is limited by the dark reactions of photosynthesis. PTOX accounts for 1% of the PSII content in *Arabidopsis thaliana*, and it is located in the non-appressed regions of thylakoids where PSI is dominant [45].

Exposing plants to high light intensities leads to an excessive reduction of the plastoquinone pool, which can cause photoinhibition. PTOX content is indicated to increase under environmental stress conditions, as shown in the C3 *Ranunculus glacialis* plant [46] exposed to full sunlight. At high light intensity, when the pH of the stroma increases, PTOX can bind to the membrane and access the substrate plastoquinol, but under nonsaturating light, the protein detaches from the membrane and is inactive [44].

It is noted that maize (NADP-ME subtype) has more PTOX in BS chloroplasts. This may be due to the agranal structure of chloroplasts and the presence of non-appressed regions. Moreover, the RuBisCO enzyme is present only in BS chloroplasts. The presence of PTOX can reduce oxygen concentration at the RuBisCO site and increase the efficiency of CO_2_ assimilation in the Calvin cycle [47]. This observation could indicate that in all metabolic subtypes of C4 plants, a higher PTOX content should be expected in chloroplasts of BS cells due to the exclusion of the oxygenase activity of RuBisCO.

There is little data on PTOX activity in individual C4 subtypes in response to high light intensities. This protein has been shown to be involved in tolerance to salt stress in a representative of PEPCK *Spartina alterniflora*, which is a halophyte. PTOX may provide an alternative pathway for the protection of this species against over-reduction and minimize or avoid damage to both photosystems (PSI and PSII) [48]. The authors suggest that *Spartina alterniflora* gained increased salt tolerance because of increased electron flow through PTOX, which can be a major sink of electrons in salt stress and functions as a safety valve to protect photosystems from over-reduction, in contrast to the response observed in the glycophyte *Setaria viridis* (NADP-ME). These experiments were carried out on whole leaves, so they do not provide information on the content and activity of PTOX individually in the M and BS chloroplasts.

The final product in the reaction catalyzed by the PTOX is water. Due to the close location of the NDH complex and the PTOX protein in the thylakoid stroma membranes, they participate in a pathway called chlororespiration, which is important in protecting PSI from over-reduction, protecting against reactive oxygen species, and preventing PSI photoinhibition [49].

### 2.3. Changes in the Amount of Thylakoid Complexes and Rearrangement of Super- and Megacomplexes

The LHCII antenna system is one of the complexes in thylakoid membranes whose content changes under conditions of varying light intensities [50]. Adaptation to high light intensities in maize is a tightly coordinated regulation of the components/activity of the light reaction in both mesophyll and bundle sheath chloroplasts [51]. Under different light intensities, both the content of individual proteins and the arrangement of the complexes can change. Low light intensity promotes the development of antenna systems to capture as much light energy as possible, in contrast to high light when LHC systems decrease. When the intensity of light increases during growth, the levels of the PSII and PSI reaction centers, as well as the cytochrome *b_6_f* complex, increase [51]. It has also been shown that the content of LHCII antenna systems increases under conditions of low light, even in maize BS chloroplasts where the PSII content is reduced.

Light, both its quantity and quality, as well as other environmental factors (e.g., temperature, CO_2_ and O_2_ concentration, drought, and also phosphate availability) affect the expression of chloroplast genes, which is dependent on the redox state of the chloroplasts. The PQH_2_/PQ ratio influences the transcription genes that encode the proteins of the PSI and PSII reaction centers [52], allowing the photosynthetic apparatus to be adjusted to the actual conditions. Maintaining the oxidized pool of plastoquinone (PQ) by exposure to PSI excitation light or by inhibiting electron transport in PSII was found to activate transcription of the *psbA* gene encoding D1—the core protein of PSII. In contrast, when the pool of plastoquinone is in a reduced state (PQH_2_), during exposure to PSII excitation light, the transcription of the *psaA* and *psaB* genes encoding the PSI reaction center proteins is activated. Variable light conditions also influence the rate of the Calvin cycle. Therefore, if the demand for ATP and NADPH and their production in light reactions of photosynthesis changes, the degree of reduction of the plastoquinone pool will also change. This has an impact on the expression of genes encoding core proteins of the photosystems, and this, in turn, may change the proportion of cyclic or linear electron transport under given conditions.

In acclimation to changing light conditions, the organization of complexes in thylakoid membranes is also important. Urban et al. [53] reported the presence of PSI–LHCI–LHCII–Lhcb4 supercomplexes and PSI–LHCI–PSII–LHCII megacomplexes in the stroma lamellae and grana margins of maize mesophyll chloroplasts. These complexes contained various LHCII trimers and monomers in combination with PSI. They were formed under both low light and high light conditions, but their exact composition differed. It was shown that exposure of plants growing in low light intensity to far-red (FR) light caused dissociation of the PSI-LHCI–LHCII–Lhcb4 supercomplex into free LHCII–Lhcb4 and PSI–LHCI complexes and which then associated with the PSII monomer. The process was different in plants grown in high light. Exposure to FR light causes dissociation of both PSI–LHCI–LHCII–Lhcb4 supercomplexes and PSI–PSII megacomplexes. These results suggest that the reorganization of the super- and megacomplex has a different function than balancing light absorption between the two photosystems under light stress. Such changes may have an influence on energy quenching and the PSII turnover cycle [53]. These studies were innovative and data on the formation of these complexes are not available for maize BS chloroplasts and for the chloroplast of other C4 subtypes.

### 2.4. Photoinhibition and Role of D1 Protein Phosphorylation

Photoinhibition is a phenomenon that leads to a decrease in photosynthetic activity and a reduction in CO_2_ assimilation. It is defined as the light-induced inhibition of photosystem II activity [54] when photosystem II degradation dominates over its repair [55]. The classic model of photoinhibition assumed the generation of reactive oxygen species by excessive reduction of the plastoquinone pool. The formed reactive oxygen species are responsible for the damage to the PSII reaction center. Currently, many authors indicate that PSII repair processes are more sensitive to environmental stresses [28,56]. Photosystem II is considered to be the most damage-sensitive complex of the thylakoid membrane, which does not mean that PSI is not affected by photoinhibition as well. PSI photoinhibition occurs when the supply of electrons from PSII exceeds the acceptor capacity of PSI [57], but PSI is effectively protected against damage, for example, by photoinhibition of a certain pool of PSII. Tikkanen et al. [58] showed that PSII photoinhibition, which reduces electron transport, prevents ROS formation and PSI damage. Furthermore, Ballotari et al. [59] observed that zeaxanthin can bind to PSI and participate in the dissipation of excess energy from PSI. It should be emphasized that the state transition process described later is also the one that protects photosystems from overexcitation. Furthermore, the results of Lima-Melo et al. [60] suggest that rapid activation of PSI photoinhibition under severe photosynthetic imbalance protects the chloroplast from over-reduction and excess ROS formation.

Repair of damaged PSII reaction centers requires the degradation of the D1 protein destroyed during photoinhibition, its de novo synthesis, and reconstruction of the PSII complex. D1 degradation is a multistage process regulated by protein phosphorylation and dephosphorylation, and also by the level of ATP in chloroplasts. The D1 protein, one of the most easily degraded, is phosphorylated under the influence of medium and high intensity of light in the granum of thylakoids by membrane-bound serine threonine kinase. This modification affects both intact and damaged reaction centers, protects against proteolytic degradation, and has no effect on the electron transport rate in PSII [61]. When irreversible damage of D1 is caused by photoinhibition, this D1 is directed to the thylakoid stroma, where it is dephosphorylated and then degraded [62].

Pokorska et al. [61] showed that the rate of degradation of the D1 protein in maize (NADP-ME plant) BS chloroplasts during photoinhibition is faster compared to mesophyll chloroplasts, and the content of Deg1 protease, which is one of the enzymes that degrades the D1 protein, was higher. Furthermore, individual steps of D1 turnover such as dephosphorylation, degradation, and de novo synthesis of PSII subunits are known to take place in stroma exposed regions, and enzymes responsible for D1 proteolysis are also present there [63]. The organization of thylakoid membranes as an agranal system in maize BS chloroplasts makes processes related to the D1 protein turnover cycle much faster.

There is no data in the literature on photoinhibition in other metabolic subtypes of C4 plants. Differences in the metabolic reaction and structure between mesophyll and bundle sheath chloroplasts may have an influence on the rate of D1 turnover.

### 2.5. State Transitions and Phosphorylation of LHCII

At different light intensities, the migration of LHCII between photosystems is observed in the process called state transitions. The LHCII antenna, and especially the Lhcb2 protein, undergoes reversible phosphorylation, which is crucial for the switching of LHCII between photosystems. The levels of LHCII phosphorylation are lower at high light compared with those under low light conditions. State 1 is traditionally defined as the condition when PSI is preferentially excited and all LHCIIs become associated with PSII. Illumination conditions, which lead to an excess excitation of photosystem II (PSII), compared to photosystem I (PSI), induce a transition to state 2, in which the more absorbed excitation energy is diverted to PSI because the phosphorylated LHCII antennas are associated with PSI [64]. State transitions act as a mechanism to balance the excitation of the two photosystems under changing light regimes [41].

The state transition has been described mainly in C3 plants, including *A. thaliana*, in which this process has been studied using mutants, e.g., in the *stn7* gene, which encodes the key enzyme, and STN7 kinase, which phosphorylates LHCII. Little is known about this process in C4 plants. In C4 plants where there are differences in the organization of thylakoid membranes in the M and BS chloroplasts, the process may be quite different. Thylakoid membranes are heterogeneous, and while PSII with the LHCII antenna is located in the stacked regions of the grana, the PSI occurs in the stroma lamellae and marginal grana regions. Thus, the number of grana in a given chloroplast type in each metabolic subtype will determine the LHCII content and the amount of PSI to which these antennas can potentially be attached.

Nakamura et al. [33] showed that with changes in the *Flaveria* genus, which led to the formation of C4 species, such as *Flaveria trinervia* (NADP-ME), increased mobility within the thylakoid membranes and increased LHCII functioning as a PSI antenna was observed in the chloroplasts of BS cells (Figure 4). It has also been shown that in maize belonging to the NADP-ME subtype, a different regulation of antenna switching is present in the two types of chloroplasts. In M chloroplasts, the state transition occurs in a classical way, and depending on the light conditions, the LHCII systems are attached to PSII or PSI. In BS chloroplasts, a permanent state 2 was observed (Figure 5), in which a certain pool of LHCII antennas remains attached to the PSI, regardless of light conditions, even at high intensities of far red, which preferentially excites PSI [41]. This can be crucial in regulating cyclic electron transport and making it work more efficiently, allowing the production of higher amounts of ATP. It should be remembered that ATP drives the energy for CO_2_ assimilation in the Calvin cycle, which operates only in BS chloroplasts.

The regulation of antenna phosphorylation in the remaining C4 subtypes may be quite different due to the organization of the thylakoid membrane in the chloroplasts and the demand for ATP. Based on this demand and the organization of the thylakoid membranes in various types of chloroplasts, we propose a possible state transition scheme (Figure 5).

### 2.6. Xanthophyll Cycle and Heat Dissipation

Among several mechanisms in chloroplasts that allow them to function under stress conditions, preventing the generation of ROS is extremely important. One of the protection mechanisms is the quenching of excess energy as a thermal dissipation. This process involves the xanthophyll cycle, related to the conversion of violaxanthin to zeaxanthin [65] and the protonation of the PsbS protein. Generally, when plants are exposed to high light intensities, violaxanthin is oxidized by violaxanthin de-epoxidase (VDE). This leads to the formation of antheraxanthin, followed by zeaxanthin. Zeaxanthin creates a barrier that prevents overexcitation of the PSII reaction center. The energy from LHCII is dissipated and is not directed to the reaction center. At low light intensities, zeaxanthin epoxidation catalyzed by zeaxanthin epoxidase (ZE) occurs. The content of carotenoids, including xanthophylls participating in the xanthophyll cycle, in M and BS chloroplasts was investigated by the group of Romanowska et al. [66]. Studies were carried out on three species of C4 plants belonging to the NADP-ME subtype: *Zea mays*, *Echinochloa crus galli*, and *Digitaria sanguinalis,* characterized by different tolerances to high light intensity. The total content of carotenoids, including zeaxanthin and antheraxanthin, a final and an intermediate product of the xanthophyll cycle, was measured. It was shown that after exposure to high light intensity, the amount of zeaxanthin in the chloroplast of barnyardgrass (*Echinochloa crus galli*) was higher than in other investigated species. Barnyardgrass is a plant highly resistant to various environmental stresses, and the functioning of the xanthophyll cycle may be one of the crucial mechanisms that allows growth under stress conditions. An increase in zeaxanthin content was also observed in *Sorghum bicolor* (NADP-ME plant), after exposure to the same light intensity as described above. The amount of zeaxanthin, antheraxanthin, and violaxanthin participating in the xanthophyll cycle was twice as high compared to the control [67], which may indicate that regardless of the species, the functioning of the xanthophyll cycle is an important element of protection of the photosynthetic apparatus and dissipation of excess energy.

Moreover, under conditions of excess light, a transthylakoidal proton gradient (ΔpH) is generated. At a low pH of the thylakoid lumen, the PsbS protein, which is involved in energy dissipation, is protonated. The dimeric forms of PsbS are associated with the PSII core, whereas the PsbS monomers are associated with the LHCII antenna [68]. Conversion of PsbS homodimers to monomers occurs at low pH in a thylakoid lumen at high light intensity [68]. The energy quenching in the form of qE is higher when there are more quenching sites, so it should depend on the PsbS protein content.

The participation of the LHCII antenna in energy quenching was confirmed in the mesophyll chloroplasts of maize (NADP-ME), where after the high intensity of far-red light, the LHCII were dephosphorylated, detached from the PSI in the stroma lamellae, moved to the grana, and either bound to PSII or formed aggregates which in consequence, lead to induction of the qE parameter [41]. In M chloroplasts, light is not a factor that limits the production of ATP and NADPH, so the excess light energy is dissipated as heat. In the agranal BS chloroplasts, there was a slight dephosphorylation of the LHCII connected to PSI, but a high dephosphorylation of free and aggregated LHCII. The dephosphorylated antenna could connect to PSII and caused an energy transfer redirecting it to this photosystem. Because the proportion of LHCII aggregates decreases, therefore, the amount of energy dissipated is also reduced [41]. The authors suggest that because of the leaf structure, less light energy can reach the BS chloroplasts, thereby decreasing the pH gradient and lowering the qE parameter.

It can be assumed that in plants that perform C4 photosynthesis (excluding those with single-cell C4 photosynthesis), the mechanisms of quenching and dissipation of excess excitation energy should be more efficient in M chloroplasts due to the specific anatomical structure of the leaves. This causes much more light energy to reach the chloroplasts localized in M cells than those in BS cells.

## 3. Summary

Mechanisms that allow C4 plants to adapt the light reaction of the photosynthesis function under changing light conditions, particularly in high light intensity, are summarized in Table 1. Many of them are universal and are also found in C3 plants. However, some are modified in C4 plants to provide more efficient CO_2_ assimilation. The close relationship between the light phase of photosynthesis and the enzymatic reactions in chloroplasts, and the associated demand for ATP and NADPH, results that in C4 plants the linear and cyclic electron transport operate in a different ratio in the chloroplasts of M and BS cells. In addition, differences in the intensity of light reaching M and BS chloroplasts and in the thylakoid structure (granal and agranal) will affect the processes of the redistribution of excitation energy between photosystems and the dissipation of its excess. Therefore, it can be assumed that, in the M chloroplasts, because of increased incoming light energy, the mechanisms related to the dissipation of excess energy must function better than in BS chloroplasts to prevent photosystems from photoinhibition and, in consequence, from a decrease in ATP and NADPH. On the other hand, BS chloroplasts, which receive less light energy, must have better functioning mechanisms that allow for its efficient use. More research is needed on other subtypes of C4 plants to help explain the importance of chloroplast structure in the processes involved in the use and dissipation of excitation energy.

## Figures and Tables

**Figure 1 ijms-23-03626-f001:**
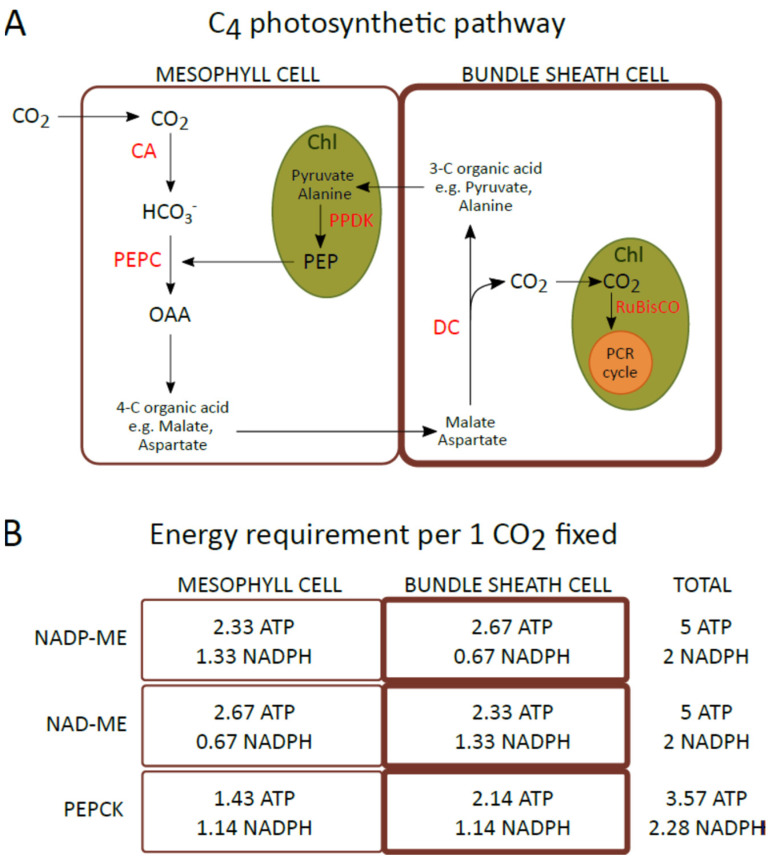
(**A**) Schematic describing the general metabolic pathway of C4 photosynthesis. (**B**) The requirement for ATP and NADPH per fixed CO_2_ in mesophyll and bundle sheath cells of plants representing the NADP-ME, NAD-ME, and PEPCK subtypes of C4 photosynthesis.ATP and NADPH requirements estimated by Edwards and Voznesenskaya [9]. CA: carbonic anhydrase; Chl: chloroplast; DC: decarboxsylase; OAA: oxaloacetate; PCR cycle: photosynthetic carbon reduction cycle; PEP: phosphoenolpyruvate; PEPC: phosphoenolpyruvate carboxylase; PPDK: pyruvate, phosphate dikinase; RuBisCO: ribulose-1,5-bisphosphate carboxylase/oxygenase.

**Figure 2 ijms-23-03626-f002:**
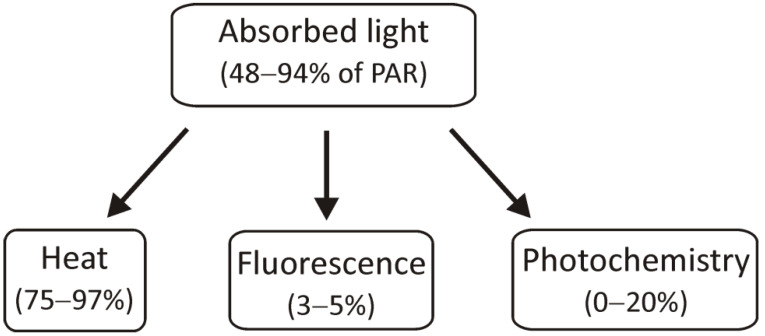
The scheme presents the use of absorbed light energy for photochemical reactions, heat dissipation, and radiation by fluorescence. PAR: photosynthetically active radiation.

**Figure 3 ijms-23-03626-f003:**
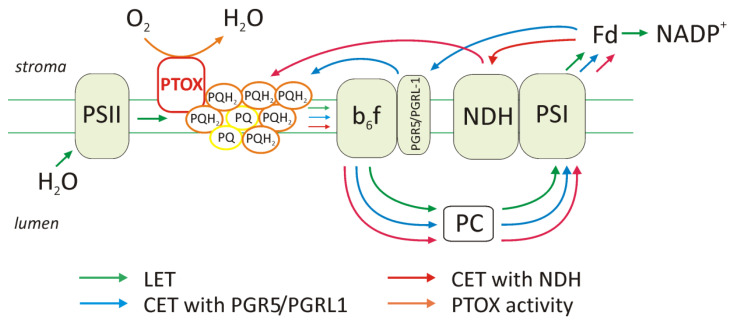
The ways of electron transport in the thylakoid membrane under high light intensities and over-reduction of the plastoquinone pool (PQ). The scheme shows linear electron transport (LET), cyclic electron transport (CET), which takes place with the participation of the proteins PGR5 and PGRL1 or/and the NDH complex, and the activity of PTOX in the chlororespiration process. The content of complexes and the activity of the electron transport pathway will differ in the M and BS chloroplasts in particular metabolic subtypes and depending on environmental conditions.

**Figure 5 ijms-23-03626-f005:**
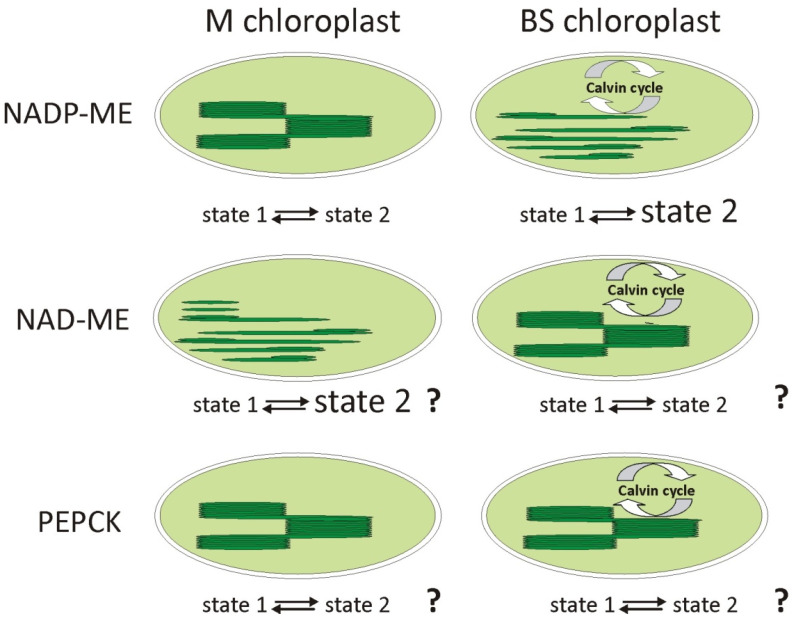
Models of state 1 state 2 transitions in two types of chloroplasts in three subtypes of C4 plants. In maize (NADP-ME) mesophyll chloroplasts, the typical transition from state 1 to state 2 occurs, while in bundle sheath chloroplasts it is permanent state 2 where some pool of LHCII antennas are bound to PSI [41]. For the NAD-ME and PEPCK subtypes, in which the regulation of antenna migration is unknown, the shown model of state transitions is based on the organization of thylakoid membranes and the demand for ATP.

**Table 1 ijms-23-03626-t001:** Summary of the chloroplast processes involved in the adaptation/acclimatization of C4 plants to high light intensities, compared to C3 plants, described in the article.

Process Taking Place in Chloroplasts	C3 Plants	C4 Plants
Xanthophyll cycle and heat dissipation	Typical, occurring with the zeaxanthin and PsbS protein [65].	
State transitions and LHCII phosphorylation	Function of state 1 and state 2, depending on phosphorylation of the LHCII antenna [64].	Permanent state 2 in agranal BS maize (NADP-ME) chloroplasts. LHCII in phosphorylated form, regardless of the condition [41].
Photoinhibition and phosphorylation of D1 protein	Damaged D1 is directed to the thylakoid stroma, dephosphorylated, and then degraded.	D1 degradation is faster in the BS chloroplast of maize [61]. Photodamage of some PSII pools for protection against PSI excess [57].
Cyclic electron transport components	Lower ATP demand resulting from metabolism.	Elevation of the CET ad alternative CET pathway with NDH complex for higher efficiency of ATP production [33,36,37].
PTOX functioning and chlororespiration	Minor importance, activity mainly under stressful conditions.	High amount and activity in maize BS chloroplasts for better protection against ROS formation during elevated cyclic electron transport [47].
Changes in antenna and reaction centers amount	Higher content of LHCII antenna in low light intensities. Higher content of reaction centers at high light intensities [51].	
Additional mechanism(s)	No data available.	Formation of megacomplexes in maize mesophyll chloroplasts [53].

## Data Availability

Data sharing not applicable.

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
