# Peer review of "How Light Reactions of Photosynthesis in C4 Plants Are Optimized and Protected under High Light Conditions"

_ijms, 2022, doi:10.3390/ijms23073626_

Round 1
Reviewer 1 Report
The authors have produced a comprehensive and readable review of their topic. It will make a valuable contribution to the special issue on Photosynthetic Reactions: From Molecules to Function, from Simple Models to Complex System.
Although the manuscript is very well written, I have made a few suggestion that will improve the expression of ideas. These are included in the attached file.

Reviewer 2 Report
The review introduces an important topic which is How light reactions of photosynthesis in C4 plants are optimized and protected under high light conditions. It is a well written review that does improve our mechanistic understanding of the mechanisms that allow C4 plants to adapt the light reaction of the photosynthesis function under changing light conditions. I also find the logic story of the sequence of the different parts of the review.
Author Response
We thank the Reviewer for their comments that we find helped us to improve the manuscript. The English has been corrected. All changes made to the manuscript were marked on the latest version using the ‘track changes’.